

# Some structural features of the peptide profile of myelin basic protein-hydrolyzing antibodies in schizophrenic patients

Maria Zavialova[1,*], Daria Kamaeva[2,*], Laura Kazieva[1], Vladlen S. Skvortsov[1] and Liudmila Smirnova[2]

[1] Institute of Biomedical Chemistry, Moscow, Russia
[2] Mental Health Research Institute, Tomsk National Research Medical Center of the Russian Academy of Sciences, Tomsk, Russia
[*] These authors contributed equally to this work.

## ABSTRACT

The antibodies of schizophrenic patients that hydrolyze myelin basic protein (MBP) have been actively studied recently, but the mechanism of the catalytic properties of immunoglobulin molecules remains unknown. Determination of specific immunoglobulin sequences associated with the high activity of MBP proteolysis will help to understand the mechanisms of abzyme catalysis. In the course of comparative mass spectrometric analysis of IgG peptides from the blood serum of patients with acute schizophrenia and healthy people, 12 sequences were identified, which were found only in antibodies that hydrolyze MBP. These sequences belong to IgG heavy chains and $\kappa$- and $\lambda$-type light chains, with eight of them belonging to variable domains. The content of peptides from the variable regions of the light chains does not correlate with the proteolytic activity of IgG to MBP in patients with schizophrenia, whereas for two sequences from the variable regions of the heavy chains (FQ(+0.98)GWVTMTR and *LYLQMN(+0.98)SLR), an increase in activity with increasing their concentration. The results suggest that these sequences may be involved in one way or another in MBP hydrolysis.

## INTRODUCTION

Numerous immunological studies in recent years show that antibodies formed in the body of mammals perform not only a receptor function, binding foreign antigens and activating the immune response, but can also exhibit catalytic activity inherent in enzymes. Such endogenous antibodies with catalytic activity are called "abzymes" (*Paul et al., 1989*). Abzymes that hydrolyze nucleic acids, polysaccharides, oligopeptides and proteins are found mainly in the blood serum of patients with autoimmune diseases, and they have also been found in patients with some viral and bacterial infections (*Nevinsky & Buneva, 2010*; *Baranovskii et al., 2001*; *Parkhomenko et al., 2014*). To date, catalytic antibodies have been found in patients with bronchial asthma (*Paul et al., 1989*; *Mei et al., 1991*), systemic

Corresponding authors
Maria Zavialova,
mariag.zavyalova@gmail.com
Daria Kamaeva, susl2008@yandex.ru

lupus erythematosus (*Bezuglova et al., 2012b*; *Paul et al., 1997*), multiple sclerosis (*Weber, Hemmer & Cepok, 2011*; *Polosukhina et al., 2005*; *Polosukhina et al., 2004*; *Niehaus et al., 2000*), Hashimoto's thyroiditis (*Paul et al., 1997*), Alzheimer's disease (*Taguchi et al., 2008*), rheumatoid arthritis (*Naumova et al., 2003*), hemophilia (*Lacroix-Desmazes et al., 1999*; *Wootla et al., 2008*).

Fractions of abzymes with proteolytic activity against polypeptides and proteins that perform key regulatory and structural functions in the body have been isolated from the blood serum of patients with various autoimmune diseases. For example, immunoglobulins G (IgGs) hydrolyzing vasoactive intestinal peptide (VIP), which is considered to mediate non-adrenergic, non-cholinergic relaxation of airway smooth muscle, have been found in patients with bronchial asthma (*Paul et al., 1989*; *Mei et al., 1991*). In the blood serum of patients with multiple sclerosis, immunoglobulins (Ig) hydrolyzing myelin basic protein (MBP) (IgG (*Polosukhina et al., 2004*; *Parshukova et al., 2020b*; *Parshukova et al., 2019*), IgA and IgM (*Polosukhina et al., 2005*)), as well as IgGs with catalytic activity to histones (*Baranova et al., 2019*) and to oligodendrocyte progenitor cell surface proteins of oligodendrocyte precursors (*Niehaus et al., 2000*) have been reported. It remains unclear whether there is a pathogenetic relationship between the effects of antibodies and the functions of abzymes in the development of the pathologies.

In schizophrenia patients, significant changes in the immune system have been observed, in particular, an increase in the level of proinflammatory cytokines and chemokines in serum, monocytosis, increased expression of inflammatory genes in monocytes and changes in the functional activity of T cells (*Bergink, Gibney & Drexhage, 2014*; *Drexhage et al., 2011*). The activation of the inflammatory response system in schizophrenia suggests the presence of neuroinflammation and potential damage to neurons and glial cells. Along with the studies identified circulating antineuronal antibodies (*Steiner et al., 2015*), this evidence may indicate underlying autoimmune and inflammatory responses in at least some patients with schizophrenia. The involvement of immune processes in the pathogenesis of schizophrenia has been suggested by many research groups (*e.g.*, see review by *Steiner et al., 2015*).

Over the past few years, group of the researchers at the Mental Health Research Institute of Tomsk National Research Medical Center of the Russian Academy of Sciences (Tomsk, Russia) in collaboration with The Institute of Chemical Biology and Fundamental Medicine of Siberian Division of Russian Academy of Sciences (Novosibirsk, Russia) has investigated a number of catalytic activities of immunoglobulins G in patients with schizophrenia. They reported that abzymes isolated from the blood serum of patients exhibited catalase activity (*Ermakov et al., 2017*), superoxide dismutase activity (*Mednova et al., 2022*), nuclease activity (*Ermakov et al., 2015*; *Ermakov et al., 2018*) and proteolytic activity to MBP (*Parshukova et al., 2020b*; *Parshukova et al., 2019*), histones (*Ermakov et al., 2020*), and collagen (*Parshukova et al., 2020a*).

Postmortem studies of schizophrenic patients using electron microscopy and histochemistry revealed structural changes in oligodendroglial cells (*Martins-de Souza, 2010*; *Martins-de Souza et al., 2009*), axonal atrophy, abnormalities of the surrounding myelin sheath (*Matthews, Eastwood & Harrison, 2012*; *Uranova et al., 2004*). In addition, a

decrease in the MBP content in the anterior frontal cortex of the brain in schizophrenia was found *Kolomeets & Uranova (2008)*, and with the help of modern quantitative methods of intravital neuroimaging, global hypomyelination of the white and gray matter of the brain was revealed in this disease (*Dietsche, Kircher & Falkenberg, 2017*; *Mighdoll et al., 2015*; *Smirnova et al., 2021*). Therefore, abzymes with proteolytic activity against neurospecific proteins, in particular, hydrolyzing MBP, have become an object of research interest.

Antibodies with proteolytic activity against MBP have been identified and characterized in detail in a number of autoimmune diseases, including multiple sclerosis, systemic lupus erythematosus (*Polosukhina et al., 2004*; *Bezuglova et al., 2011*), as well as in autism (*Gonzalez-Gronow et al., 2015*), schizophrenia (*Parshukova et al., 2020b*; *Parshukova et al., 2019*) and bipolar disorder (*Baranova et al., 2019*). Nevertheless, their unambiguous pathogenetic role and the structural features that provide the catalytic properties of antibodies are currently unknown. There is still no unambiguous understanding of which structural parts of antibody proteins are responsible for their catalytic activity. IgG antibodies are hypervariable globular proteins of Y-shape consisted of two heavy and two light chains interconnected by disulfide bonds. *Paul et al. (1989)* first reported that polypeptides and proteins are hydrolyzed not only by whole catalytic antibodies, but also by their Fab-fragments, as well as isolated light chains (*Mei et al., 1991*).

*Bezuglova et al. (2012b)* showed that in patients with systemic lupus erythematosus, IgG of all four subclasses (IgG1–IgG4) exhibit proteolytic activity against MBP including their $\lambda$- and $\kappa$-type light chains (*Bezuglova et al., 2012b*). At the same time, an increase in catalytic activity was observed in line IgG4 < IgG2 < IgG3 < IgG1 and IgGs containing $\lambda$-light chains demonstrated a higher relative MBP-hydrolysing activity than $\kappa$-IgG. The participation of light chains in the implementation of the IgG's catalytic activity has been shown in a number of studies on natural and recombinant antibodies with different activities (*Bezuglova et al., 2011*; *Gonzalez-Gronow et al., 2015*; *Kamaeva et al., 2022*; *Sapparapu et al., 2012*; *Hifumi et al., 2020*; *Faber & Whitehead, 2019*). Another work by Bezuglova showed the proteolytic activity of serum IgG's light chains in patients with multiple sclerosis (*Bezuglova et al., 2012a*). The next work of the same group of authors described the proteolytic activity of 72 recombinant monoclonal $\kappa$-type light chains cloned from patients with systemic lupus erythematosus (*Timofeeva, Buneva & Nevinsky, 2015*). The results indicated that 22 of the 72 IgG's monoclonal light chains hydrolyzed MBP specifically. Proteolytic activity of these light chains is characterized by varying pH-optimum, sensitivity to inhibitors (PMSF, EDTA and iodoacetamide (IAA)) and dependence from metal cations. According to the results of inhibitory analysis, four light $\kappa$-chains exhibited serine-protease-type activity, three of them having thiol-protease activity, and eleven light chains showed metal-dependence and were classified as metalloproteinases (*Timofeeva, Buneva & Nevinsky, 2015*). Thus, natural polyclonal abzymes specific for one antigen can be very diverse in their catalytic properties. The purpose of this work was to determine the common peptide sets of IgGs (peptide profile) that are specific for abzymes exhibiting high proteolytic activity against MBP. We assume that the exploration of such peptide profile will make it possible to distinguish between IgGs with different proteolytic activities.

## MATERIALS & METHODS

### Subjects

Blood serum samples from eight schizophrenia patients (five females and three males) and six healthy individuals (three females and three males) were used as the objects for the study. The patients were admitted for inpatient treatment at the clinic of the Mental Health Research Institute of the Tomsk National Research Medical Center of the Russian Academy of Sciences (Tomsk NRMC). Diagnostic assessment and clinical qualifications were performed by psychiatrists in accordance with ICD-10. Six patients were diagnosed with paranoid schizophrenia (F20.0) and two of them had simple schizophrenia (F20.6). All of included patients demonstrated predominant negative symptoms. The average age of the patients was $36.71 \pm 14.94$ years, the duration of the disease was $10.86 \pm 9.54$ years. The control group consisted of mentally and somatically healthy individuals matched by sex and age ($31.67 \pm 6.2$ years). Exclusion criteria for all participants were the following: presence of acute or chronic infections in the last 2 months, inflammatory, autoimmune or neurological diseases, other organic mental disorders and mental retardation. All individuals included in the study gave written informed consent. This study was conducted according to the guidelines of the Declaration of Helsinki and was approved by the Biomedicine Ethic Committee of the Tomsk National Research Medical Center of the Russian Academy of Sciences (protocol number 147/4.2021, date of approval 29 June 2021).

### Study samples

Blood serum samples were collected as previously described in *Parshukova et al. (2020b)*. Specifically blood samples were obtained after overnight fast from a vein into tubes with a clot activator (CAT, BD Vacutainer, Franklin Lakes, NJ, USA). To isolate the serum, the blood samples were centrifuged for 30 min at $2,000 \times g$ at 4 °C. The sera were stored at −80 °C until analysis.

### Purification of IgGs

Immunoglobulins G were isolated from blood serum by affinity chromatography on AKTA Pure chromatography system GE Healthcare on a Protein G Sepharose column (*Polosukhina et al., 2004*; *Smirnova et al., 2021*). Serum diluted 1:3 with buffer A (50 mM Tris–HCl, pH 7.5, 150 mM NaCl) was loaded onto the Protein G Sepharose column. Proteins that did not interact with the sorbent were washed with 8 ml of the same buffer until the absorption at 280 nm completely disappeared. Nonspecifically adsorbed proteins and lipids were eluted with 3 ml of buffer A containing 1% Triton X-100 and 0.3 M NaCl. Then the column was washed with 15 ml of buffer A. IgGs were eluted with 0.1 M glycine-HCl, pH 2.6. The isolated antibody fractions were neutralized with 1 M Tris–HCl, pH 8.8. After isolation, all IgG samples were subjected to homogeneity analysis using 5–18% SDS-PAGE and Coomassie staining.

### Measuring of proteolytic activity of IgGs

The proteolytic activity of antibodies to MBP was measured according to the previously described method (*Parshukova et al., 2020b*). The reaction mixture with a volume of 10 µl

contained: 20 mM Tris–HCl, pH 7.5; 20 mM NaCl, 0.1 mg/ml of MBP and antibodies at a concentration of 0.2 mg/ml. The reaction mixture was incubated for 24 h at a temperature of 37 °C, after which the reaction products were analyzed by SDS electrophoresis in 15% PAGE followed by Coomassie staining, as described previously (*Parshukova et al., 2020b*). Gels were visualized using the iBright Imaging Systems FL1500 gel documentation system (Thermo Fisher Scientific, Waltham, MA, USA) on base of The Core Facility "Medical Genomics" Tomsk NMRC. The analysis of the efficiency of MBP hydrolysis by antibodies was carried out according to the decrease in the intensity of MBP band after incubation with individual samples of IgG relative to the color intensity of MBP after incubation without IgG using the ImageQuant 5.2 software.

## Sample preparation of IgG samples for mass spectrometry

The protein concentrations in IgG samples were measured by the bicinchoninic acid (BCA) reaction method using a commercial BCA protein assay kit according to the manufacturer's protocol (Pierce, Appleton, WI, USA).

Samples of IgG were prepared for mass spectrometric analysis as follows. Aliquots of antibody samples containing 25 μg of protein were dialyzed on centrifuge filters (Microcon 30K; Millipore, Burlington, MA, USA). For this, 400 μl of 100 mM Tris, pH 8.5, with 0.2% sodium deoxycholate were added to the samples and centrifuged for 8 min at 14,000 × g. The washing procedure was repeated three times, then the solutions above the filter were transferred into clean test tubes, and 20 μl of 1.5 M thiourea was added to the obtained antibody samples (∼50 μl).

Proteins were reduced by adding 5 μl of 300 mM DTT in 0.1 M ammonium bicarbonate. After incubation at 56°C for 30 min, 5 μl of 300 mM IAA in 0.1 M ammonium bicarbonate was added. The reaction mixture was incubated in dark for 30 min at room temperature. To quench the remaining IAA, 5 μl of 300 mM DTT in 0.1 M ammonium bicarbonate was added. The sample was diluted up to 200 μl with 0.1 M ammonium bicarbonate. Trypsin was added in enzyme/protein ratio 1:50. Digestion was performed at 37 °C overnight. After digestion, formic acid was added to the samples to the final concentration of 1%. Samples were centrifuged at 10 °C at 12,000 × g for 10 min. Supernatants were collected, dried under vacuum and resuspended in 20 mkl of 0.1% formic acid in MilliQ water.

## Mass spectrometry analysis

High-performance liquid chromatography and mass spectrometry analysis was performed similarly to *Rusanov et al. (2021)* with minimal changes.

The tryptic peptides were separated with high-performance liquid chromatography using Ultimate 3000 Nano LC System (Thermo Fisher Scientific, Waltham, MA, USA). One microliter of sample, equal to 1 microgram of peptides, was loaded directly onto the C18 column (Acclaim® PepMap™ RSLC, 75 μm × 15 cm, Thermo Fisher Scientific, Waltham, MA, USA) at a flow rate of 0.3 μl/min for 12 min in an isocratic mode of Mobile Phase C (2% acetonitrile, 0.1% formic acid). Then peptides were eluted with a gradient of buffer B (80% acetonitrile, 0.1% formic acid) at a flow rate of 0.3 μl/min. Total run time was 110 min, which included initial 12 min of column equilibration to buffer A (0.1%

formic acid), then gradient from 5 to 55% of buffer B over 75 min, then 6 min to reach 99% of buffer B, flushing 10 min with 99% of buffer B and 7 min re-equilibration to buffer A.

Mass spectrometry (MS) analysis was performed with a Q Exactive HF-X mass spectrometer (Thermo Fisher Scientific, Waltham, MA, USA). The temperature of capillary was 240 °C and the voltage at the emitter was 2.1 kV. Samples were analyzed in triplicate in Full MS mode followed by single data dependent MS2 (DDA MS2). Mass spectra were acquired in a mass range of 320–1,500 m/z with a resolution setting 120,000 (MS). Precursor ions were fragmented in HCD (high-energy collisional dissociation) mode. Tandem mass spectra of fragments were acquired at a resolution of 15,000 (MS/MS) in the range from 140 m/z to the m/z value determined by a charge state of the precursor, but no more than 2,000 m/z. The maximum accumulation time of precursor and fragment ions was 50 ms and 100 ms, correspondently. The AGC target for precursor and fragment ions were set to $1 \times 10^6$ and $2 \times 10^5$, correspondingly. An isolation intensity threshold of 50,000 counts was determined for the selection of precursors and up to top 20 precursors were chosen for fragmentation at 30 NCE. The isolation width for precursor ions was 2 m/z. Precursors with a charge state of 1+ and more than 5+ were rejected, and all measured precursors were dynamically excluded from triggering of a subsequent MS/MS for 20 s (*Rusanov et al., 2021*).

## Proteomic analysis

Protein sequences of the complete human proteome provided by UniprotKB (August 2021) was used for protein identification with MaxQuant (using the Andromeda search engine). Protein identification was conducted against a concatenated target/decoy version. Potential contaminants were contained in the sequences database and they were identified using common contaminants list built into MaxQuant. Carbamidomethylation of cysteines was set as a fixed modification and N/Q deamidation as well as oxidation of methionine residues was set as variable modifications for the peptide search. A maximum m/z deviation of 5 ppm was allowed for the identification of the precursor, and 10 ppm was set as match tolerance for fragment identification. One missed cleavage was allowed for tryptic digestion. The false discovery rates (FDR) for peptide and protein identifications were set to 1%. To increase the reliability of identifications, features were aligned between MS runs within a time window of 2 min by the software option "Match between runs". According to the criteria defined by The Human Proteome Organization (HUPO), proteins were considered as reliably identified if at least two peptides were identified for them (*Carr et al., 2004*).

## Quantitative and statistical analyses

Protein label-free quantification was performed by MaxQuant. Unique and razor peptides were used for label-free quantification with iBAQ algorithm. We determined relative molar abundances for each protein in the sample similarly to *Krey et al. (2014)* by its relative value riBAQ (riBAQ $=$ iBAQ$_{protein}$/ $\sum$iBAQ$_{all\ proteins}$).

Peptide label-free quantification was performed by Progenesis LC-MS (Nonlinear Dynamics, Milford, MA, USA). The obtained raw files were processed using Progenesis

LC-MS. It was used for sequential alignment of experimental LC-MS runs of the samples and counting of ion abundances for detected m/z features.

The data were statistically analyzed in Progenesis QI, software for LC-MS data analysis. Within Progenesis QI, a one-way analysis of variance (ANOVA) test was used to assess significance between groups (control *vs* schizophrenia) and returned a *P*-value for each feature. ANOVA (p) ≤0.05 was taken as significant by default. Additionally, Principal Component Analysis (PCA) was used in this work. PCA in Progenesis QI uses compound abundance levels across runs to determine the principle axes of abundance variation. Transforming and plotting the abundance data in principle component space separates the run samples according to abundance variation.

### Target analysis

The most abundant, previously unidentified peptide ions were subjected to targeted mass spectrometric analysis. We modified previously described LC-MS/MS by adding of the inclusion list of m/z values for data independent analysis (DIA). The selected ions were isolated with 2 m/z isolation width and fragmented with HCD with sequentially varying NCE value: 24%, 27%, and 30%. Fragmentation spectra were analyzed using the PEAKS Studio (BSI, Ontario, Canada) using a sequence library composed of sequences of immunoglobulin light and heavy chains from the GenBank database. The search for the corresponding amino acid sequences and *de novo* sequencing were carried out with mass tolerance settings 5 ppm for the m/z precursor, and 0.01 Da for m/z fragments, taking into account the variable modifications of methionine oxidation (M), cysteine carbamidomethylation (C), asparagine/glutamine deamidation (N/Q). The maximum number of variable post-translational modifications per peptide is 10. The predicted *de novo* sequences were compared using BLAST with human protein sequences using the UniProtKB database to determine whether they belong to class G immunoglobulins. Refined sequence variants were manually validated using the Peptide fragmentation modeling utility (PFM) in the Molecular Weight Calculator (PNAS) by comparing the measured and theoretical spectra of fragments with ion-matching window of 0.01 Da.

## RESULTS

The purification of immunoglobulins was performed individually from serum samples of donors using Protein G-Sepharose sorbent. The elution profiles of IgG with acidic buffer represented homogeneous peaks of optical density, which corresponded to the profile of immunoglobulin concentration in the samples. At the first step of this work, the homogeneity of the antibodies was analyzed using 5–18% SDS-PAGE. SDS-electrophoretic analysis of IgG samples obtained by affinity chromatography under non-reducing conditions led to the identification of protein bands with a molecular weight of 150 kDa, which corresponds to IgG (Fig. 1). The other colored bands corresponding to proteins are not visualized, indicating the absence of contamination by co-eluted products.

For further analysis, fourteen IgG samples were selected including eight samples from schizophrenia patients with proteolytic activity to MBP and six samples from healthy individuals with negligible or "null" proteolytic activity (Table 1). The median level of

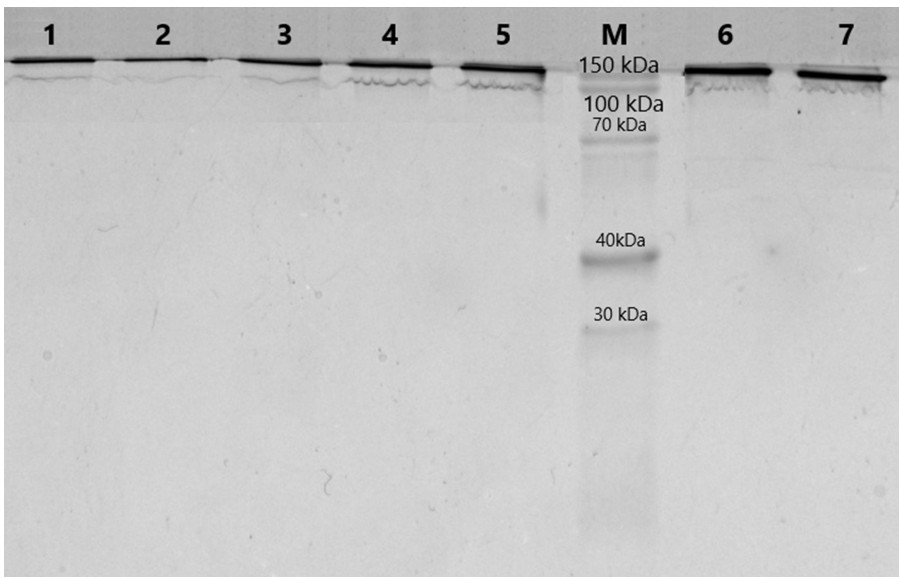

**Figure 1  Analysis of homogeneity of IgGs after 5–18% SDS-PAGE and Coomassie staining.** Lanes 1–7, IgG of different schizophrenia patients; M, protein molecular mass markers.

**Table 1  Values of proteolytic activity of IgG to MBP in patients with schizophrenia and healthy volunteers (control).**

| Sample number | Group | Proteolytic activity, mg MBP/mg IgG/h |
|---|---|---|
| C1 | Control | 0.02 |
| C2 | Control | 0 |
| C3 | Control | 0.02 |
| C4 | Control | 0.01 |
| C5 | Control | 0 |
| C6 | Control | 0.1 |
| SCH1 | Schizophrenia | 1.21 |
| SCH2 | Schizophrenia | 1.16 |
| SCH3 | Schizophrenia | 1.51 |
| SCH4 | Schizophrenia | 1.39 |
| SCH5 | Schizophrenia | 5 |
| SCH6 | Schizophrenia | 0.41 |
| SCH7 | Schizophrenia | 1.01 |
| SCH8 | Schizophrenia | 0.71 |

MBP-hydrolyzing activity in the group of patients was 1.18 [0.86; 1.45] mg MBP/ mg IgG/ h, which significantly exceeded the level of activity in the group of healthy individuals 0.015 [0.00; 0.02] mg MBP/ mg IgG/ h ($p = 0.002$, Mann–Whitney test).

As a result of proteomic analysis using MaxQuant software, 169 proteins were identified in the submitted samples, 20 of which were classified as contaminants always detected in mass spectrometry-based proteomics, such as keratins, and 86 were plasma proteins
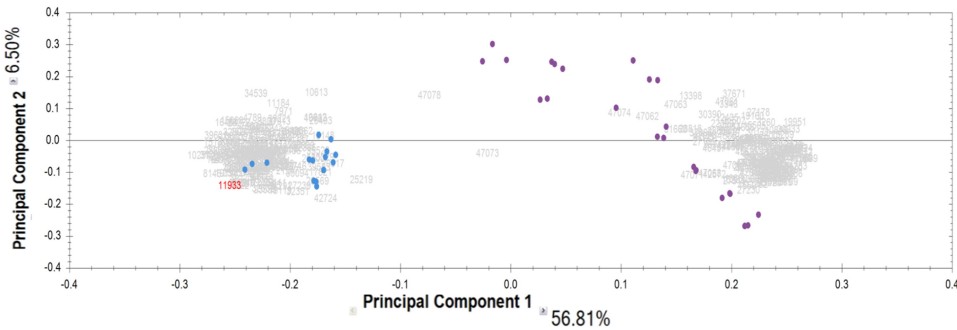

**Figure 2** Results of principal component analysis (PCA) by IgG groups: blue dots—samples of the control group; purple dots—samples of patients with schizophrenia.

that did not belong to the IgGs. The list of identified proteins is presented in Table S1. We did not reveal IgG sequences that were unique to all patients with schizophrenia. Based on the values of the relative molar content of riBAQ proteins, also shown in Table S1, it can be seen that the total abundance of immunoglobulins in the samples averages 92.4% ($\sigma = \pm 2.8\%$), and the rest of the proteins are impurities and are identified due to the high sensitivity of mass spectrometry. Among the contaminating proteins were five proteins that were present in some samples of MBP-hydrolyzing IgGs and were not found in control samples: Complement C1q subcomponent subunit C (five samples from eight), Ceruloplasmin (three samples), Plasminogen (four samples), Inter-alpha-trypsin inhibitor heavy chain H2 (two samples), Vitamin D-binding protein (four samples). The data on the molecular functions of the identified proteins obtained using the GeneOntology (GO) knowledge bases are shown in Table S1.

Proteomic analysis of the IgG samples tryptic digests with the following analysis of peptide LC-MS profiles using Progenesis revealed a total of 13,798 peptide ions (features). To identify patterns of specific IgG peptides, differing in groups with high and low proteolytic activity against MBP, we filtered out features that met the following criteria: (1) fold change > 5; (2) ANOVA ($p$) < 0.00001. As a result of a comparative analysis of antibody samples with high proteolytic activity (from patients) and with low activity (control group), we identified 321 peptide ions (features) specific for a particular group. Among them, 160 ions were most represented in the control group, and 161 ions were most represented in the group of patients with a high level of MBP hydrolysis. According to the PCA results for the selected ions, the significant difference between IgG samples with proteolytic activity and the control group is about 57% for the first component (Fig. 2). At the same time, we did not observe any difference between the subgroups of patients with paranoid and simple schizophrenia.

A minor part of the peptides that determine the differences in the groups belonged to major plasma proteins identified in shotgun mass spectrometry analysis. In the antibody samples with increased MBP-hydrolyzing abzyme activity we observed an increased amount of peptide alpha 2-macroglobulin (LHTEAQIQEEGTVVELTGR,

LPPNVVEESAR), serotransferrin (APNHAVVTR, IECVSAETTEDCIAK, LCMGS-GLNLCEPNNK, EGTCPEAPTDECKPVK), haptoglobin (HYEGSTVPEKK, GSFP-WQAK), complement component C1q (FQSVFTVTR, TINVPLRR) and albumin (NEC(+57)FLQHKDDNPNLPR, EVQLVESGGR, AAC(+57)LLPK, LVAASQAALG, SLHTLFGDKLCTVATLR, LKECCEKPLLEK, pyro-QEPERNEC(+57)FLQHK, pyro-QEPERNECPNLQHK). Hereinafter, value in the parentheses in the peptide sequence indicates a peptide mass shift due to the peptide modification. These proteins are well studied and do not have proteolytic properties against MBP according to their molecular functions (Table S1).

We focused our attention on unidentified peptide ions from the samples specific for MBP-hydrolyzing antibodies because their sequence may contain amino acid patterns that determine the proteolytic activity of anti-MBP abzymes. We selected 112 high-intensity MS ions that were only detectable in MBP-hydrolyzing antibodies and estimated the dependence of the measured proteolytic activity of antibodies on the relative abundance of the selected peptides in the samples, which were determined from the area of the XIC peak on the corresponding peptide ion (Table S2). We observed an upward trend for 17 ion features that were detected in at least three IgG samples in all technical replicates. Graphically, the dependence of the MBP hydrolysis activity on the XIC value of the selected ions is shown in Suplementary materials (Fig. S1). Ion abundances were increased in accordance with proteolytic activity with an approximation reliability of R2 more than 0.9. Our task was to decrypt as many sequences of MS features specific for IgG samples with high proteolytic activity as possible because such specific sequences may not be the same but only similar in BLAST alignment. We performed the targeted MS/MS of the selected 112 ions in pooled antibody samples followed by *de novo* sequencing. From the predicted variants of the peptides, we selected those that had the greatest coincidence with the immunoglobulin sequences using BLAST alignment. The established full or partial peptide sequences and their corresponding immunoglobulins are shown in Table 2. Details of *De novo* are presented in Suplementary materials (Fig. S2).

Among the peptides that are found only in IgG samples with proteolytic activity against MBP, the amino acid sequences of 4 peptides from the constant domains of the heavy chains of immunoglobulins were determined, as well as five and three peptides in the variable regions of the heavy and light chains, respectively. Since, according to the literature, it is the light chains of immunoglobulins that exhibit proteolytic function, first of all it is worth considering the identified sequences of light chains specific for the MBP-hydrolyzing antibodies group. The three predicted sequences belong to the variable region V1 of the λ- or κ-chain, one sequence of QRPSGVPDR corresponds to the V3 region, but it also has high homology with another NRPSGIPDR sequence from the V1 region proposed for m/z = $338.177^{3+}$. Thus, it can be assumed that the proteolytic activity of abzymes can be regulated precisely in the variable region V1.

Based on the results of the MS measurements (Table S2), we see that the hydrolyzing activity value does not depend on the abundance of m/z = $338.177^{3+}$ and m/z = $851.393^{2+}$, and for m/z = $749.869^{2+}$ this dependence cannot be estimated because this peptide was detected in only two samples. A similar assessment of the proteolytic activity of

**Table 2  List of immunoglobulin peptides identified in PEAKS belonging to IgG from schizophrenia patients.**

| m/z | z+ | MW | Sequence | Score DB (−10lgP)/ de novo (%) | Protein | RT, min | Number of samples with m/z (total $N = 8$) |
|---|---|---|---|---|---|---|---|
| 338.177 | 3 | 1011.51 | N(+0.98)RPSGIPDR | 17.9/NA | Ig lambda variable V3 | 24.6 | 5 |
| | | | Q(+0.98)RPSGVPDR | 16.6/NA | Ig lambda variable V1/V6 | | |
| 427.219 | 3 | 1278.636 | SVLHQDWLN(−17)GK | NA/59 | IGHG | 48.6 | 6 |
| 430.219 | 4 | 1716.847 | SVLHQDWLN(+0.98)GKEYK | 18.2/NA | IGHG | 49.2 | 2 |
| | | | SVLHQN(+0.98)WLDGKEYK | 18.2/NA | | | |
| | | | TVVHQDWLN(+0.98)GKEYK | 15.8/NA | | | |
| | | | TVVHQN(+0.98)WLDGKEYK | 15.8/NA | | | |
| 467.742 | 2 | 933.47 | LSC(+57)AASGLR | 20.9/NA | IGHV3 | 26.9 | 4 |
| 490.733 | 2 | 979.452 | HQDWLN(−17)GK | 38.5/NA | IGHG | 38.4 | 1 |
| 494.596 | 3 | 1480.767 | NVN(+0.98)HKPSNTKVDK | 23.5/NA | IGHG | 12.1 | 2 |
| 552.29 | 3 | 1653.848 | EVQLVESGGGLEQPGR | 35.6/NA | IGHV3 | 46.9 | 5 |
| 556.254 | 2 | 1110.494 | TNYADSVEGR | 28.9/NA | IGHV3 | 31.8 | 1 |
| 563.78 | 2 | 1125.545 | FQ(+0.98)GWVTMTR | 30.9/NA | IGHV1-2 | 62.3 | 5 |
| 613.3 | 3 | 1836.878 | * LYLQMN(+0.98)SLR | 21.5/NA | IGHV3 | 73.8 | 4 |
| 749.869 | 2 | 1497.724 | VTLSC(+57)SGTYANLGR | NA/63 | Ig lambda variable V1 | 50.5 | 2 |
| 851.393 | 2 | 1700.772 | SLQPEDVATYYC(+57)QK | NA/68 | Ig kappa variable (1–27) | 59.3 | 4 |

**Notes.**
DB, database match; IGHG, Immunoglobulin heavy constant gamma; IGHV, Immunoglobulin heavy variable; NA, not available data.
Value in the parentheses in the peptide sequence indicates a peptide mass shift due to the peptide modification.

antibodies against the identified peptides of the variable domains of IgG heavy chains showed different results. The peptide content of LSC(+57)AASGLR (m/z = $467.742^{2+}$), EVQLVESGGGLEQPGR (m/z = $552.29^{3+}$) and TNYADSVEGR (m/z = $556.254^{2+}$) does not correlate with the activity of abzymes, while peptide FQ(+0.98)GWVTMTR (m/z = $563.78^{2+}$) and a peptide with a partially decoded sequence *LYLQMN(+0.98)SLR (m/z = $613.3^{3+}$) show an upward trend.

Six of 17 features with an upward trend were fully or partially sequenced by PEAKS. Two sequences were aligned to IgG. The FQ(+0.98)GWVTMTR peptide belongs to the IgG variable domain 1-2, and its sequence coincides with 100% coverage with a single protein Immunoglobulin heavy variable 1-2 (P23083 according to the UniProtKB database). Peptide *LYLQMN(+0.98)SLR (m/z = $613.3^{3+}$) is in the third variable domain because the LYLQMNSLR sequence in BLAST alignment coincides 100% with 18 sequences of the heavy chain V3 variable regions. Figure 3 shows the correspondence of b- and y-fragments of the *de novo* predicted peptides to the peaks in the MS/MS spectra. Peptide ion m/z = $613.3^{3+}$ is partially decoded by fragments. Measured molecular mass does not match the proposed sequence. The C-terminal sequence *LYLQMN(+0.98)SLR is confirmed by good quality fragments, but the exact order of residues SL or LS is not defined and determined from BLAST alignment, D and deamidated N are also not distinguishable by mass shift.

Proteolytic activity increased with increasing abundance in four out of five IgG samples for m/z = $563.78^{2+}$ where it was detected, and similarly in four out of four IgG samples

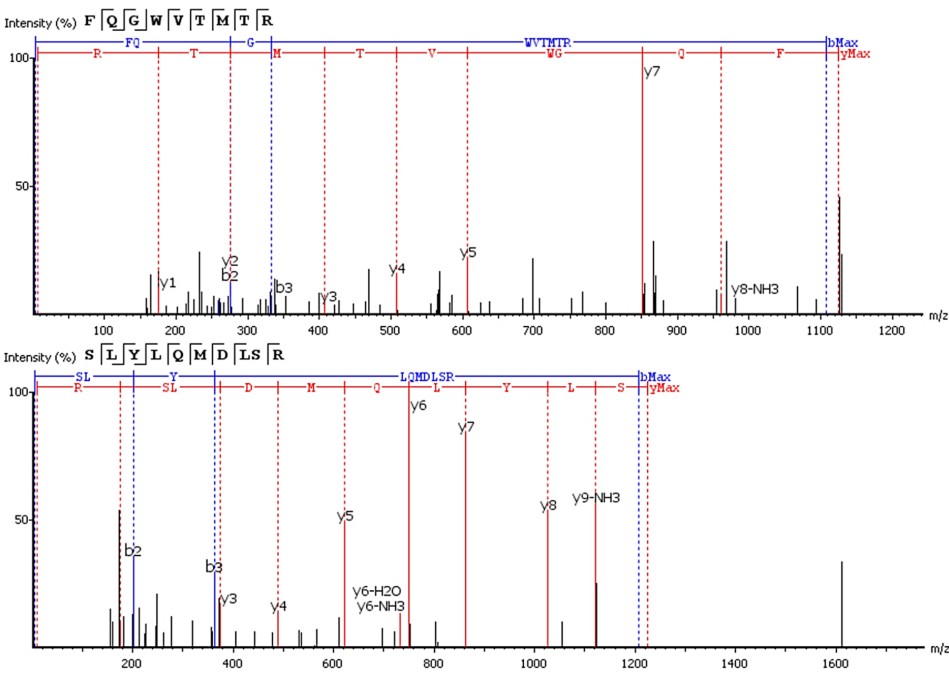

**Figure 3** *De novo* sequencing of peptides from HCD MS/MS: top spectrum—FQ(+0.98)GWVTMTR in MS/MS spectrum of m/z = 563.78$^{2+}$; (B) bottom spectrum—*LYLQMN(+0.98)SLR in MS/MS spectrum of m/z = 613.3$^{3+}$.

for m/z = 613.3$^{3+}$. Combination of both peptides was found in three samples of IgG-hydrolyzing MBP. Graphically, the dependence of the MBP hydrolysis activity on the XIC value of the selected ions is shown in Fig. 4.

## DISCUSSION

Structural studies of immunoglobulins are currently being actively developed because of their high importance for biotechnology and biomedical research. Nevertheless, mass spectrometric analysis of immunoglobulins G in mental disorders is currently sporadic in the literature. In this article, we have described for the first time some structural features of the peptide profile of MBP-hydrolyzing antibodies in schizophrenic patients revealed using bottom-up proteomic analysis.

According to modern concepts, the assignment of catalytic activity directly to purified from biological samples natural abzymes requires verification of a number of stringent criteria (*Paul et al., 1989*; *Nevinsky & Buneva, 2010*; *Bezuglova et al., 2011*). The main, usually tested criteria include: purification of IgGs by affinity chromatography, electrophoretic homogeneity of the immunoglobulins, maintaining the catalytic activity of antibodies during their gel filtration under "pH shock" conditions and detection of enzymatic activity *in situ* in the gel regions occupied by immunoglobulin bands, their Fab-fragments, or individual chains. The reproducibility of this approach has been shown in studies by various groups of authors over the past 30 years, including MBP-hydrolyzing

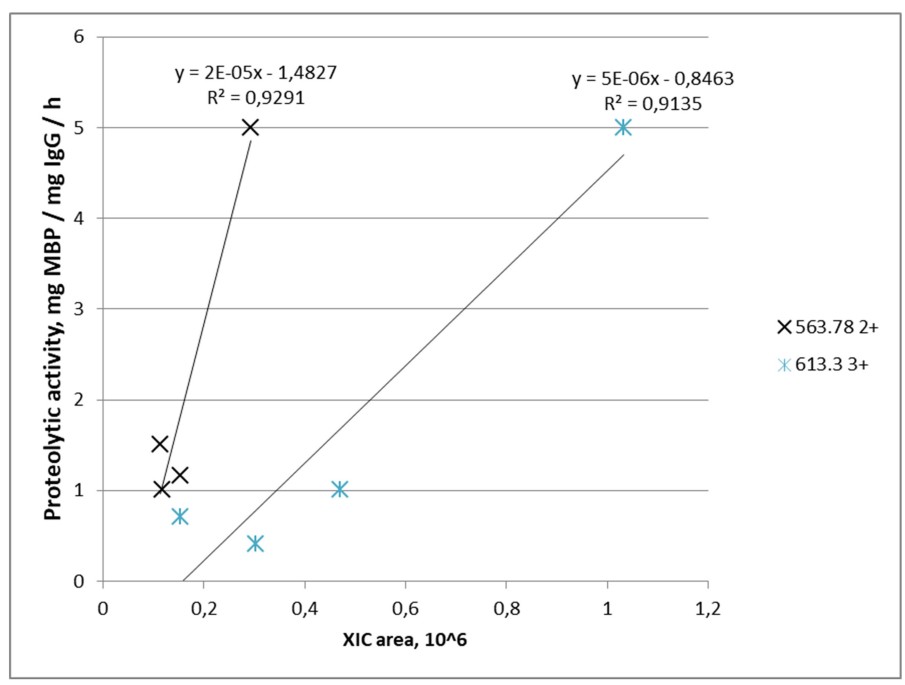

**Figure 4 Dependence of proteolytic activity of IgG against MBP on the abundance of target peptides: a cross—for the FQ(+0.98)GWVTMTR peptide; an asterisk—for the *LYLQMN(+0.98)SLR peptide.** Each data point indicates the correspondence of measured integral MS signal of peptide ion (XIC value), which is proportional to peptide abundance in the IgG sample (*x*-axis), and measured proteolytic activity of this IgG (*y*-axis). A cross determined the FQ(+0.98)GWVTMTR peptide; an asterisk determined the *LYLQMN(+0.98)SLR peptide.

IgGs in schizophrenia (*Polosukhina et al., 2005*; *Polosukhina et al., 2004*; *Parshukova et al., 2020b*; *Parshukova et al., 2019*; *Timofeeva, Buneva & Nevinsky, 2015*). In our study, we used antibodies purified by affinity chromatography, and the homogeneity of the analyzed samples was shown using SDS-PAGE. However, in subsequent proteomic analysis using MaxQuant software, 20 proteins were classified as contaminants that are mainly present in the laboratory environment and are always detected in proteomic experiments, such as keratins, and 86 were plasma proteins that did not belong to the IgGs. Based on the values of the relative molar content of riBAQ proteins, it can be seen that the purity of immunoglobulins has reached more than 92%, and the rest of the proteins are impurities from plasma proteins, that were identified due to the high sensitivity of mass spectrometry. Shotgun MS is capable to detect proteins at a concentration of $10^{-9}$ M–$10^{-12}$ M (*Vavilov et al., 2022*). Standard Coomassie staining, used in the work, allows visualization from 100 ng of protein in a SDS-PAGE spot (*Weiss, Weiland & Görg, 2009*), which exceeds the detection limits of mass spectrometry by 2–5 orders of magnitude in terms of protein amount. Protein G affinity chromatography is a widely used and effective method for the purification of immunoglobulins. G protein has a high affinity for the Fc regions from G-class immunoglobulins, which allows selective elution of components of the immune complexes (proteins, polysaccharides, nucleic acids) under conditions with increased ionic

strength or in the presence of nonionic detergents without destroying the Ig complexes with protein G (*Grodzki & Berenstein, 2010*). However, this method does not completely avoid the co-eluting of proteins and, according to the manufacturer's instructions, small amount of major serum proteins may be isolated as contaminants. Nevertheless, we consider the detected proteolytic activity as an intrinsic property of antibodies. Among contaminating proteins, we identified five proteins that were present in some samples of MBP-hydrolyzing IgGs and were not found in control samples: Complement C1q subcomponent subunit C, Ceruloplasmin, Plasminogen, Inter-alpha-trypsin inhibitor heavy chain H2, Vitamin D-binding protein. These proteins were observed in less than 2/3 of the samples with MBP-hydrolyzing activity, *i.e.,* the proteolytic activity was detected regardless of the presence of these contaminating proteins. Proteomic analysis of peptides from the IgG samples using Progenesis revealed an increased amount of peptide groups belonged to major plasma alpha 2-macroglobulin serotransferrin, haptoglobin and albumin. These proteins do not have proteolytic properties according to GO (Table S1), except for haptoglobin. Despite being homologous to serine proteases, haptoglobin has lost all essential catalytic residues and has no enzymatic activity (*Kurosky et al., 1980*).

In our study, we tried to determine specific peptide profiles of immunoglobulins, indicating a high proteolytic activity of IgG to MBP, and to identify their amino acid sequences using *de novo* mass spectrometric sequencing. MS analysis of the polyclonal IgG samples identified patterns of specific IgG peptides, differing in groups with high proteolytic activity and low proteolytic activity against MBP consisted 321 peptide ions specific for a particular group. Among the peptides that are found only in IgG samples with proteolytic activity against MBP we identified 12 sequences belonging to IgG heavy chains and $\kappa$- and $\lambda$-type light chains, eight of them were from variable regions and of four peptides from the constant domains of the heavy chains. A large number of characteristic peptides determining differences in groups were not identified at bottom-up proteomic analysis even if they have high quality MS/MS spectra. It is highly probable that these peptides belong to the variable domains of antibodies, and their sequences do not coincide with the reference ones from the UniProt database. Since we used polyclonal serum IgG in this study, this result indicates a significant difference in the IgG repertoire between patients with schizophrenia and healthy individuals. There is some evidence that the prevalence of antibodies to various microbial and endogenous antigens increased in subset of patients with schizophrenia in comparison with healthy individuals (*Ermakov et al., 2022*). Mass spectrometric studies in vaccination, infection, and autoimmune diseases have shown that the repertoire of circulating immunoglobulins is dominated by a limited number of antibody clones (*Snapkov et al., 2022*). It is noted that many antibody clones persist over years, but the proportion of each is changing, thus similar overall autoantibody titers may give variable disease activity (*Chen et al., 2017*).

Catalytic properties, according to studies of mutagenesis, biochemistry, genetics and crystallography, are associated with the variable (V) domain of immunoglobulins (*Gao et al., 1995*; *Sapparapu et al., 2012*; *Le Minoux et al., 2012*; *Ramsland et al., 2006*). While C-domain is an important factor influencing expression of V-domain catalytic activity (*Sapparapu et al., 2012*). The V domains contain conformation-dependent

nucleophilic sites that are fully competent in completing the first step in catalytic hydrolysis of amide and peptide bonds, nucleophilic attack on electrophilic carbonyl groups (*Sapparapu et al., 2012*; *Smirnov et al., 2011*). The V-domains of immunoglobulins expressing the nucleophilic catalytic sites are encoded by genes displaying a high conservation degree with germline genes thus showing that catalysis is an innate antibody function (*Le Minoux et al., 2012*; *Gololobov, Sun & Paul, 1999*). The variable domain of IgG is formed by a light chain and a heavy chain. Each of the variable heavy (VH) or light (VL) domains contains three hypervariable regions, which together constitute six complementarity determining regions (CDRs) that form the surface of the antigen-binding site of an antibody or paratope (*Chailyan, Marcatili & Tramontano, 2011*; *Janeway Jr et al., 2001*). Intact proteins usually interact with both light and heavy chains of abzymes, thereby providing specificity in the recognition of the target protein and its cleavage.

The three predicted sequences belong to the light chains variable region V1 of the λ- or κ-chain, one sequence corresponds to the V3 region, but it also has high homology with another sequence from the V1 region proposed. Thus, it can be assumed that the proteolytic activity of abzymes can be regulated precisely in the variable region V1. According to most studies essential catalytic residues forming the catalytic center are usually located in the VL domain and additional residues from the VH domain are involved in high affinity binding of the substrate (*Mei et al., 1991*; *Timofeeva, Buneva & Nevinsky, 2015*; *Sun et al., 1995*; *Sun et al., 1997*). As shown earlier by *Timofeeva, Buneva & Nevinsky (2015)*, monoclonal light chains of antibodies that hydrolyze MBP differ greatly in their proteolytic properties. However, in our study, the quantity of the peptides from the light chain variable domains in IgGs from patients with schizophrenia does not correlate with the proteolytic activity against MBP. At the same time an increase of concentration of two peptides—FQ(+0.98)GWVTMTR from the heavy chain variable domain 1–2 (also CDR1 of heavy chain) (m/z = $563.78^{2+}$) and *LYLQMN(+0.98)SLR (m/z = $613.3^{3+}$) from heavy chain variable regions 3 (also CDR3 of heavy chain)—correlate with the an increase in proteolytic activity of antibodies. As can be seen from Table 2, these peptides were found in five and four samples, respectively, among eight studied samples with a pronounced MBP-hydrolyzing activity. In the case of MBP proteolysis by natural abzymes there is probably observed a cumulative effect both from different polyclonal antibodies and from the combination of different variable and constant regions in one monoclone (*Timofeeva, Buneva & Nevinsky, 2015*). Therefore, the measured proteolytic activity can multifactorially dependent on various light chain peptides. The identified sequences can be part of the sites of antigenic recognition and binding of MBP, together with variable regions of heavy chains (*Chailyan, Marcatili & Tramontano, 2011*).

Some studies show that inter-molecular interactions between heavy and light chains, variable and constant regions of immunoglobulin can play an important role in the overall molecular structure of paratope in relationship to its affinity and specificity for antigen (*Janda et al., 2016*). Also VL and VH domains may have mutual effects, in particular due to relative spatial positions in the six complementarity determining regions in V domains structural diversity of CDR3 of heavy chain might influence the conformation of CDR3 of light chain (*Kuroda et al., 2009*). In addition, crystallography of natural proteolytic

antibody identified a putative catalytic triad in V domain formed with participation of residues from both light and heavy chains (*Ramsland et al., 2006*). *Kim et al. (2006)*, in a study of DNA hydrolysis activity, also indicates the presence of catalytic properties in isolated heavy chain variable domains VH.

Taking into account the above studies, we can assume that the identified sequences are part of variable domains of heavy chain that exhibit affinity specificity for MBP and potentially impact effectiveness of catalysis, but it is still impossible to draw final conclusions about which sites are involved in MBP hydrolysis.

## CONCLUSIONS

Antibodies with proteolytic activity against myelin basic protein have been identified and characterized in detail in a number of autoimmune diseases, including multiple sclerosis, systemic lupus erythematosus, as well as in autism and schizophrenia. Nevertheless, their unambiguous pathogenetic role, as well as the structural features that provide the catalytic properties of antibodies in schizophrenia, have not been characterized. To date, proteomic analysis of specific IgG peptide sequences is the only method that allows one to determine and characterize the variable regions of natural antibodies responsible for the presence of catalytic properties.

In our study, we tried to determine specific peptide profiles of immunoglobulins, indicating a high proteolytic activity of IgG to MBP, and to identify their amino acid sequences using *de novo* mass spectrometric sequencing. As a result, 12 sequences belonging to IgG heavy chains and $\kappa$- and $\lambda$-type light chains were identified in MBP-hydrolyzing abzymes, eight of them were from variable regions. In our study, the quantity of the peptides from the light chain variable domains in IgGs from patients with schizophrenia does not correlate with the proteolytic activity against MBP, whereas for two peptides from the heavy chain variable regions FQ(+0.98)GWVTMTR(m/z $= 563.78^{2+}$) and *LYLQMN(+0.98)SLR (m/z $= 613.3^{3+}$) there is an increase in activity with an increase in their concentration. Based on these results we can assume that these sequences are involved in MBP hydrolysis. This pilot proteomic study was performed on a small group of IgG samples from schizophrenic patients and healthy individuals, which is a limitation of the study. But to clarify the mechanisms of catalytic activity and the role of the identified peptides, more extensive research is needed. Thus, the study of the content of variable domains of light and heavy chains of IgG possessing catalytic activity will make it possible to understand the mechanisms of this activity. In addition, the study of the relationship of certain variable domains with catalytic activity may, in the future, have prognostic value.

## ACKNOWLEDGEMENTS

Mass spectrometric measurements were performed using the equipment of "Human Proteome" Core Facility of the Institute of Biomedical Chemistry (Russia, Moscow).

### Funding
This research was funded by the Russian Scientific Foundation grant No 21-75-00071 (a part of work corresponding to collection of the sera of schizophrenia patients, purification of antibodies and determination of proteolytic activity was done). The proteomic experiments were done in the framework of the Russian Federation fundamental research program for the long-term period for 2021–2030 (No 122030100168-2). The funders had no role in study design, data collection and analysis, decision to publish, or preparation of the manuscript.

### Grant Disclosures
The following grant information was disclosed by the authors:
Russian Scientific Foundation: 21-75-00071.
Russian Federation fundamental research: 122030100168-2.

### Competing Interests
The authors declare there are no competing interests.

### Author Contributions
- Maria Zavialova conceived and designed the experiments, performed the experiments, analyzed the data, prepared figures and/or tables, authored or reviewed drafts of the article, and approved the final draft.
- Daria Kamaeva conceived and designed the experiments, performed the experiments, analyzed the data, prepared figures and/or tables, authored or reviewed drafts of the article, and approved the final draft.
- Laura Kazieva performed the experiments, prepared figures and/or tables, and approved the final draft.
- Vladlen S. Skvortsov analyzed the data, prepared figures and/or tables, and approved the final draft.
- Liudmila Smirnova conceived and designed the experiments, authored or reviewed drafts of the article, supervision, and approved the final draft.

### Human Ethics
The following information was supplied relating to ethical approvals (*i.e.*, approving body and any reference numbers):

This study was conducted according to the guidelines of the Declaration of Helsinki and was approved by the Biomedicine Ethic Committee of the Tomsk National Research Medical Center of the Russian Academy of Sciences (protocol number 147/4.2021, date of approval 29 June 2021).

### Data Availability
The raw data is available at Figshare: Zavialova, Maria; Kamaeva, Daria; Kazieva, Laura; Skvortsov, Vladlen; Smirnova, Liudmila (2023): Some structural features of the peptide

profile of myelin basic protein-hydrolyzing antibodies in schizophrenic patients. figshare. Dataset. https://doi.org/10.6084/m9.figshare.21995621.v2.

## Supplemental Information

Supplemental information for this article can be found online at http://dx.doi.org/10.7717/peerj.15584#supplemental-information.

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
