# Peer review of "Some structural features of the peptide profile of myelin basic protein-hydrolyzing antibodies in schizophrenic patients"

_PeerJ, doi:10.7717/peerj.15584_

## Round 0.1 · original submission · Minor Revisions

I have now received the reviewers' comments on your manuscript. They have suggested some minor revisions to your manuscript. Therefore, I invite you to respond to the reviewers' comments and revise your manuscript.

·

Basic reporting

Dear Editor,
I really appreciate the opportunity to review the manuscript peerj-81896 entitled:
"Some structural features of the peptide proûle of myelin basic protein-hydrolyzing antibodies in schizophrenic patients"

I commend the authors for describing this critical and timely issue. The paper is interesting and well-written; however, I would like to highlight some issues that merit revision:

The article comes across as innovative, really well drafted, and written.

Experimental design

My only observation concerns the composition of the sample under study. Although it is understandable how the in-depth genetic analysis carried out is very difficult to conduct on large samples, it is equally important to point this out by adding a short line indicating it as a limitation.

Validity of the findings

It should also be specified why a case-control was chosen, or at least that is the way it was posed, without proceeding with a strict pairing by classes (e.g., age, gender), with a ratio of 1:2, or even better 1:3 ta cases and healthy controls.
I beg the authors to briefly describe this aspect (2.3 lines will suffice) in the manuscript in order to enable the reader to have well in mind the relative weight of the results in relation to the sample.

Additional comments

Overall, a very good job. Kudos to the authors

Reviewer 2 ·

Basic reporting

I thank the authors for the interesting results they present in this article. Due to the high sensitivity of the mass spectrometry method, it is possible to identify impurities in isolated igG preparations. However, despite the detected impurities, in this work it was shown that it is the blood antibodies of patients with schizophrenia that are active in MBP hydrolysis. In this work, for the first time, a mass spectrometry method was proposed to identify the regions of antibodies responsible for the presence of catalytic properties. Minor remarks (listed below) must be corrected prior to Acceptance.

I suggest that you improve the description at:

Lines 56-59: relationship between pro-inflammatory cytokines, chemokines, etc. . with antibodies discussed in this paper is not clear.

Lines 104-106: what do you mean by peptide profile? What is the novelty of this work?

Line 141: please indicate homogeneity testing of IgG preparations
Lines 148-149: indicate SDS-PAGE staining

Line 249. The description of the results should begin with the characterization of donors and IgG isolation procedures.
Line 258. Please complete with a description of the results of the analysis of the activity of MBP hydrolysis with antibodies from patients with schizophrenia in comparison with control donors.

Line 264-266 and 383-384: “20 proteins were classified as contaminants”, but these proteins are also plasma proteins. It also states that 86 plasma proteins have been identified. Please explain the number of identified proteins.

The Figures and Tables should be improved in the following ways:
Fig.1. poor separation of samples in the gel. Please extend the SDS-PAGE time.
The caption to Fig. 2 should be described in more detail. Why are the results for only four antibodies shown? What do these equations show?
Fig.3. Please check the caption for the figure: b)?
Fig.4. The picture is not clear. The figure caption should be more detailed.
Table 1. Please indicate the number of independent repetitions of these experiments. Give the standard deviation.

Experimental design

no comment

Validity of the findings

no comment

·

Basic reporting

The manuscript by Zavialova and coworkers titled “Some structural features of the peptide profile of myelin basic protein-hydrolyzing antibodies in schizophrenic patients” describes comparison of mass spectrometry-based peptide profiles of serum immunoglobulins of schizophrenic patients and healthy subjects. The authors identified two peptides that are likely to possess catalytic ability to hydrolyze myelin basic protein. This is an excellent study, and the study design is appropriate for the research question. The introduction is written well, and the methods were clearly described. Results were clearly explained in the text and adequately supported by the figures. The discussion was appropriately written with some exceptions.
Please address the following issues.
1) Please describe how contaminants were identified? Provide detailed methodological details.
2) There were 20 contaminants (Table s1). Authors claim that these contaminants are detected due to high sensitivity of mass spectrometry. I think this statement is misleading. Many of these contaminants have more peptides detected than some immunoglobulins. In fact, a large portion of contaminants are keratins which are present in ambient lab environment. Therefore, keratins are always detected in mass spectrometry-based proteomics. Please include this in the manuscript.
3) In figure 2, please identify paranoid and simple schizophrenia in different colors. What are the grey colored numbers in figure 2? What is their significance?

Experimental design

-

Validity of the findings

-

---

## Round 0.2 · accepted · Accept

In my opinion, this manuscript has been revised with attention to the reviewers' comments and can now be published.

Reviewer 2 ·

Basic reporting

The new additions to the manuscript made a big difference. The quality of the paper had improved, and all my questions were addressed. No more comments.

Experimental design

The new additions to the manuscript made a big difference. The quality of the paper had improved, and all my questions were addressed. No more comments.

Validity of the findings

The new additions to the manuscript made a big difference. The quality of the paper had improved, and all my questions were addressed. No more comments.

Additional comments

The new additions to the manuscript made a big difference. The quality of the paper had improved, and all my questions were addressed. No more comments.

·

Basic reporting

The authors addressed my comments.

Experimental design

-

Validity of the findings

-